# Comparative Structure–Activity Analysis of the Antimicrobial Activity, Cytotoxicity, and Mechanism of Action of the Fungal Cyclohexadepsipeptides Enniatins and Beauvericin

**DOI:** 10.3390/toxins11090514

**Published:** 2019-09-03

**Authors:** Hamza Olleik, Cendrine Nicoletti, Mickael Lafond, Elise Courvoisier-Dezord, Peiwen Xue, Akram Hijazi, Elias Baydoun, Josette Perrier, Marc Maresca

**Affiliations:** 1Centrale Marseille, CNRS, iSm2 UMR 7313, Aix Marseille University, 13397 Marseille, France (C.N.) (M.L.) (E.C.-D.) (P.X.) (J.P.); 2Department of Biology, American University of Beirut, Beirut 1107 2020, Lebanon; 3Doctoral School of Science and Technology, PRASE, Lebanese University, Beirut 5, Lebanon

**Keywords:** cyclic fungal peptides, cyclohexadepsipeptide, enniatin, beauvericin, mycotoxin, antibiotic, antimicrobial peptide, AMP, *Clostridium perfringens*

## Abstract

Filamentous fungi, although producing noxious molecules such as mycotoxins, have been used to produce numerous drugs active against human diseases such as paclitaxel, statins, and penicillin, saving millions of human lives. Cyclodepsipeptides are fungal molecules with potentially adverse and positive effects. Although these peptides are not novel, comparative studies of their antimicrobial activity, toxicity, and mechanism of action are still to be identified. In this study, the fungal cyclohexadepsipeptides enniatin (ENN) and beauvericin (BEA) were assessed to determine their antimicrobial activity and cytotoxicity against human cells. Results showed that these peptides were active against Gram-positive bacteria, *Mycobacterium,* and fungi, but not against Gram-negative bacteria. ENN and BEA had a limited hemolytic effect, yet were found to be toxic at low doses to nucleated human cells. Both peptides also interacted with bacterial lipids, causing low to no membrane permeabilization, but induced membrane depolarization and inhibition of macromolecules synthesis. The structure–activity analysis showed that the chemical nature of the side chains present on ENN and BEA (either *iso*-propyl, *sec*-butyl, or phenylmethyl) impacts their interaction with lipids, antimicrobial action, and toxicity.

## 1. Introduction

Bacterial resistance is considered as a major public health concern, particularly when considering hospitals where nosocomial infections are prevalent in intensive care units [1,2]. The World Health Organization proposed that “we are heading towards a post-antibiotic era in which common infections and minor injuries can once again kill”. Undeniably, antibiotic resistance is rising worldwide, threatening our ability to cure common infectious diseases. More troublesome infections lead to higher medical costs, prolonged hospital stays, and increased mortality. For these reasons, it is important to find novel antibiotics to lower the chance of acquired resistance. Antimicrobial peptides (AMPs) are promising solutions to eradicate bacterial infections, since their mechanism of action, mostly based on targeting the bacterial membrane, differs from classical antibiotics that are targeting key intracellular enzymes and/or macromolecule synthesis. AMPs are not only active against bacteria resistant to classical antibiotics, but are also less prompt at inducing/selecting mutants resistant to them. Reassuringly, very few AMP resistance cases have been reported [1].

Fungi produce a large variety of cyclic peptides, including cyclic fungal peptides with antimicrobial activity, such as cyclodepsipeptides. These peptides are produced by bacteria, plants, algae, sponges, and other marine organisms [3]. Cyclodepsipeptides, also known as peptolides, are cyclooligomers, biosynthesized by non-ribosomal peptide synthetases (NRPS) in combination with either polyketide synthase (PKS) or fatty acid (FA) synthase enzyme systems [3]. In cyclodepsipeptides, one or more amino acid is replaced by a hydroxylated carboxylic acid, resulting in the formation of at least one lactone bond in the core ring. Enniatins (ENN), including ENN A, A1, B, and B1 (Figure 1) are fungal cyclohexadepsipeptides with six residues that alternate between N-methyl amino acids and hydroxylated carboxylic acids. ENN are produced by different fungal species, such as *Verticillium*, *Halosarpheia,* and *Fusarium* species (such as *Fusarium subglutinans, Fusarium proliferatum,* or *Fusarium avenaceum*) that are plant pathogens, mycotoxins producers, and opportunistic human pathogens [4,5,6]. ENN A was the first of this group to be discovered 70 years ago and currently, 29 ENN have been reported, with expectations of new analogues to be available in the future [4]. Beauvericin (BEA) is another fungal cyclohexadepsipeptide with a core structure made of three N-methyl-L-phenylalanine units connected alternately with three 2-hydroxy-D-isovaleric acid residues (Figure 1). BEA is produced by many entomophathogenic fungi such as *Beauveria*, *Isaria*, *Cordyceps,* and *Fusarium* species [3].

ENN and BEA are referred to as mycotoxins due to their known toxic properties [7,8,9,10]. The Food and Agriculture Organization of the United Nations estimated that approximately 25% of the cereals produced in the world are contaminated by mycotoxins [11]. Exposure to mycotoxins leads to alterations of the health and it is referred to as mycotoxicosis [11]. Mycotoxins have been identified as etiological factors in different human diseases and are able to cause physiological alterations such as feed refusal, diminished resistance to infectious agents in animals, immunosuppression, carcinogenicity, neurotoxicity, nephrotoxicity, teratogenicity, as well as reproductive and developmental toxicity [5,12,13]. These effects led many countries to adopt regulations to limit mycotoxin exposure, for example in 2012, mycotoxins were the main hazard in border rejection notifications in the European Union [11]. 

ENN and BEA usually contaminate grains like wheat, barley, maize, and rice. Contamination levels of ENN and BEA have been studied worldwide with results ranging from trace-level up to 520 mg/kg [5]. Due to their lipophilic nature, ENN and BEA may bioaccumulate in different organs or tissues, entering into the food chain through meat, milk, or eggs [11]. Other foods, like nuts, spices, fruits, and their by-products can also be contaminated by these fungal metabolites [11]. In terms of toxicity, ENN and BEA have been shown to be toxic to different animals and humans cell types causing apoptosis, mitochondrial damages, and reactive oxygen species (ROS) production [7,8,10,14,15].

Alternatively, studies showed the potential of ENN and BEA as candidates for therapeutic application. For example, ENN were shown to have antibacterial activity against different bacterial species [16]. Additionally, a mixture of ENN, called fusafungine, was approved for topical treatment of upper respiratory tract infections [17]. Other studies showed that ENN may also have antifungal activity [18], anti-HIV activity [19], and antiparasitic activity against helminthes [20]. BEA is reported to possess antifungal [21] and antibacterial activity [22]. Furthermore, BEA has been shown to be a chemosensitizing agent by inhibiting the efflux of antibiotics leading to increased antibiotic effectiveness [23]. In addition, both ENN and BEA have anticancer properties [24,25]. Finally, they have hypolipidemic effects related to their ability to inhibit Acyl coenzyme A cholesterol acyltransferase (ACAT) and triglyceride biosynthesis [26].

Overall, the literature provides examples of both deleterious and beneficial effects of ENN and BEA. It is difficult to determine whether these molecules should be considered as toxins or medicines, particularly as these effects were determined independently in different publications, using different cell models, bacterial strains, assays, doses, and source/purity of ENN or BEA. To answer this question, in the present study, a comparative structure–activity analysis of the antimicrobial and cytotoxic activities of the main ENN (ENN A, A1, B, B1) and BEA was conducted to determine the active doses for each activity. In addition, the mechanism of action of ENN and BEA explaining their antibacterial action was studied.

## 2. Results

### 2.1. ENN and BEA Possess Antibacterial and Antifungal Activities

Antibacterial activity of ENN and BEA was tested against a panel of Gram-positive and Gram-negative bacteria (Table 1) by determining their minimal inhibitory concentrations (MIC) as explained in Materials and Methods. Regarding antibacterial activity, results showed that ENN and BEA were active against Gram-positive bacterial species and *Mycobacterium* with no activity against tested Gram-negative strains. In all cases, ENN A was found the more active, ENN B being the less active with an MIC > 100 µM for all tested Gram-positive bacteria except *Clostridium perfringens*. 

### 2.2. ENN and BEA Have Low Hemolytic Activity

Hemolytic activity of ENN and BEA was evaluated using human erythrocytes, as described in Materials and Methods section. Although all peptides showed hemolytic activity in a concentration-dependent manner, their hemolytic activity was found to be limited and dependent on the peptide considered (Figure 2 and Table 2). Thus, the peptide with the highest hemolytic activity was ENN A followed by BEA, ENN A1, B1, and B with HC_50_ (i.e., the concentration of peptide causing 50% of hemolysis) of 68.7 ± 9.6, 118.9 ± 15.8, 162.7 ± 22.8, 356.4 ± 59.3, and 534.5 ± 86.5 µM, respectively.

### 2.3. ENN and BEA are Toxic to Nucleated Human Cells

Cytotoxicity assays were completed on BEAS-2B (human normal airway cells), Caco-2 (human intestinal cell line), HEK (human normal keratinocytes), HEPG2 (human liver cell line), HUVEC (human normal vascular endothelial cells), and N87 (human gastric cell line) cells as explained in Materials and Methods section. After 48 h of exposure, all the peptides caused a dose-dependent reduction of the cell viability (Figure 3). The inhibitory concentrations 50 (IC_50_) (i.e., the concentration of peptide reducing by 50% the cell viability) were calculated from Figure 3 and are given in Table 2. Results showed that N87 cells, corresponding to human gastric cell line, are particularly sensitive to ENN and BEA toxicity with IC_50_ ranging from 0.003 to 27.5 µM for ENN A1 and BEA, respectively, and with the following order of efficiency: ENN A1 > B1 > A >> B >> BEA. 

For HUVEC cells (normal human endothelial cells), IC_50_ ranged from 2.4 to 17.3 µM with the following order: BEA > A > A1 > B1 > B. For HEPG2 (human liver cell line), IC_50_ ranged from 3.0 to 5.6 µM with the following order: A > B = BEA > A1 = B1. For the other cell type tested (i.e., BEAS-2B, Caco-2, and HEK), ENN A and A1 were found the most toxic (with IC_50_ ranging from 1.1 to 5.7 µM, 2.3 to 6.4 µM, and 2.4 to 6.3 µM, respectively), BEA, ENN B, and B1 being found the less toxic (with IC_50_ ranging from 3.9 to 6.3 µM, 4.6 to 54.2 µM, and 2.7 to 12.7 µM, respectively). Interestingly, BEAS-2B cells corresponding to human normal airway epithelial cells were found to be less sensitive to ENN and BEA toxicity with IC_50_ ranging from 5.7 to 43.7 µM, potentially explaining the safe use of these molecules to treat infections of the upper respiratory tract through topical treatment [17].

### 2.4. ENN and BEA Interact and Insert into Bacterial Lipids

Most AMPs act through interaction and permeabilization of the bacterial membrane. Similarly, an insertion of ENN and BEA into the membrane may explain their cytotoxic and hemolytic activities. For that reason, the insertion of ENN and BEA into bacterial or eukaryotic lipids was measured using lipid monolayers. Lipids tested correspond to the ones present on the outer leaflet of the eukaryotic or bacterial membranes. On one hand, phosphatidyl-choline (POPC) is only present in the outer membrane of eukaryotic cells. On the other hand, phosphatidyl-glycerol (POPG), phosphatidyl-ethanolamine (POPE), cardiolipin, and lipoteichoic acid (LTA) are present in the outer membrane of Gram-positive bacteria. The efficiency of insertion of cyclohexadepsipeptides into lipid monolayers was evaluated by measuring their critical pressure of insertion as explained in Materials and Methods. The critical pressure of insertion of each peptide into a particular lipid was calculated as the theoretical value of initial pressure of lipid monolayer not permissive to peptide insertion, i.e., causing a variation of pressure equal to 0 mN/m. Critical pressure of insertion is a very important parameter reflecting the ability of a peptide to insert into the lipid monolayer, its value increasing for peptide with higher insertion capacity. Values of critical pressures of insertion of ENN and BEA into eukaryotic or bacterial lipids were calculated from Figure 4 and are given in Table 3. Results showed that lipid selectivity dependents on the peptide considered. For ENN A, A1, and B1, the order of lipid selectivity was LTA > cardiolipin > POPG > POPC > POPE. For ENN B, the order of lipid selectivity was very close with LTA > POPG > cardiolipin > POPC > POPE. The higher critical pressure of insertion was obtained with LTA for all ENN in the following order ENN A > A1 >> B1 > B in accordance with the higher activity of these peptides towards Gram-positive bacteria and with the MIC results showing that ENN A and A1 are more active than ENN B1 or B. Conversely, BEA only inserted into POPC and POPE monolayer with the following order of selectivity POPC > POPE and did no insert into LTA, POPG, or cardiolipin. Interestingly, in accordance with their hemolytic activity (Figure 2 and Table 2), the critical pressure of insertion of the peptides in POPC (the main lipid found in the outer membrane leaflet of the red blood cells) was found to be BEA > ENN A > A1 > B1 > B (with critical pressure of insertion of 41.65, 40.28, 38.06, 36.19, and 32.92 mN/m, respectively).

### 2.5. ENN and BEA Permeabilize the Bacterial Membrane but Only at High Doses

Having shown that the cyclohexadepsipeptides ENN and BEA could insert into bacterial lipids, their effect on the integrity of the bacterial membrane was evaluated using propidium iodide assay as explained in Materials and Methods section (Figure 5, Figure 6 and Figure 7). 

*C. perfringens* was used in this assay and the other mechanistic assays as it is the only tested Gram-positive strain found sensitive to BEA and all ENN, including ENN B that is inactive on *S. aureus* and *B. subtilis*. At their MIC (Figure 5) or four times their MIC values (Figure 6), ENN and BEA caused low to no permeabilization of bacteria with percentage of permeabilization at four times the MIC ranging from 1.8 ± 2.8% to 7.2 ± 2.5% for BEA and ENN A, respectively. Longer incubation times with high concentrations of peptides (i.e., 120 min with 100 µM of ENN or BEA) resulted in a moderate membrane permeabilization with percentages ranging from 19.6 ± 7.9% to 37.1 ± 6.7% for ENN B and ENN A, respectively (Figure 7).

### 2.6. ENN and BEA at Low Doses Cause the Depolarization of the Bacterial Membrane

Bacterial membrane permeabilization assay demonstrated that, although they possess antibacterial effect against Gram-positive bacteria and are able to insert into bacterial lipids, ENN and BEA caused low to no bacterial permeabilization. This suggests that, contrary to most known AMPs, ENN and BEA act through an alternative mechanism of action to kill bacteria. Previous studies have demonstrated that ENN and BEA can act as ionophores on artificial membrane (i.e., liposomes) [17,27,28,29,30] and eukaryotic membrane [31,32], but their effect on the bacterial membrane potential has never been addressed or described. The effect of ENN and BEA on the membrane potential was tested using *C. perfringens* as model bacteria. All peptides caused a dose-dependent membrane depolarization of *C. perfringens* (Figure 8) in an order reflecting their MIC values, i.e., with the following order of efficiency: ENN A > A1 > B1 > BEA > B with respectively 108.9 ± 7.3%, 97.4 ± 5.3%, 97.6 ± 9.1%, 59.1 ± 2.0%, 44.9 ± 8.7% membrane depolarization at 100 µM. At lower concentrations of peptides corresponding to their MIC (Figure 9) or four times their MIC (Figure 10), ENN and BEA also caused a significant membrane depolarization of *C. perfringens*, with the following order of efficiency: ENN A > A1 > B1 > BEA > B with respectively 55.3 ± 4.5%, 50.1 ± 7.3%, 47.4 ± 4.6%, 38.3 ± 3.3%, 36.4 ± 3.8% and 80.8 ± 3.2%, 75.3 ± 2.0%, 68.0 ± 4.3%, 63.8 ± 4.8%, 44.2 ±1.8% depolarization at their MIC or four times their MIC, respectively.

### 2.7. ENN and BEA at Low Doses Inhibit the Synthesis of Bacterial Macromolecules

Specific fluorescent staining of bacteria followed by microscopic observation was successfully used by others and us as an alternative technique to radioactivity to evaluate the effect of antibiotics on bacterial macromolecule synthesis [33,34]. This technique relies on the comparison of the morphological changes caused by test compounds and conventional antibiotics with known mechanism of action used as controls. *C. perfringens* was exposed to ENN, BEA, or conventional antibiotics at five times their MIC values for 4 h before labelling of their membrane using FM 4–64 (red fluorescence) or their DNA using DAPI (blue fluorescence) and observation of the bacterial morphology under microscope. Results showed that conventional antibiotic, ENN and BEA caused particular morphological phenotypes showed in Figure 11, Figure 12 and Figure 13 and listed in Table 4. Thus, Gemifloxacin (targeting DNA synthesis) and Amoxicillin (targeting cell wall biosynthesis) caused both an elongation of the cells (5- to 6-times increase in the cell length), but differed in their effect on the condensation of DNA, i.e., only Gemifloxacin causing the presence of decondensed DNA. Rifampicin (targeting RNA synthesis) caused DNA condensation in one filament crossing the cells with low to no bacterial elongation. ENN A and A1 caused the same phenotype, showing that these two peptides could similarly target the RNA synthesis. On another hand, like Tetracycline (targeting protein synthesis), ENN B, B1, and BEA caused the formation of 1–4 DNA clusters inside the cells with low to no cell elongation demonstrating that these peptides also inhibit the protein synthesis in *C. perfringens*.

## 3. Discussion

Antimicrobial resistance is a large-scale problem; antimicrobial peptides (AMPs) are active against antibiotic-resistant strains and they could be excellent candidates as future treatments. Cyclodepsipeptides are cyclooligomers produced by non-ribosomal peptide synthetases (NRPS) in combination with either polyketide synthase (PKS) or fatty acid (FA) synthase enzyme systems [3]. These peptides have been shown to possess various activities, including antibacterial, antifungal, antiviral, anticancer, and others [3,35]. In the present study, the antimicrobial activity, the cytotoxicity, and the mechanism of action of the fungal cyclohexadepsipeptides ENN and BEA were studied. Regarding their antibacterial effect, results showed that ENN and BEA are active against *Mycobacterium* and Gram-positive bacteria, particularly against *C. perfringens*, but not against Gram-negative ones. Importantly, ENN and BEA were found active against Gram-positive bacteria resistant to conventional antibiotics such as Nisin, Vancomycin, or Methicillin. Results also showed that the activity of the different isoforms of ENN and BEA are not equivalent. On *B. subtilis* and *S. aureus*, the following order of efficiency was found: ENN A = BEA > A1 > B1 > B, whereas on *C. perfringens*, *E. faecalis,* and *M. smegmatis*, the order was: ENN A = A1 > B1 = BEA > B, ENN A > A1 > B1 = BEA > B, and ENN A1 > A = B1 > BEA > B, respectively. In all cases, ENN B was found to be less active with a MIC > 100 µM for all Gram-positive bacteria tested except *C. perfringens* that was the only Gram-positive bacterial strain tested sensitive to it with an MIC value of 12.5 µM. Regarding antifungal activity, ENN and BEA were found active on *C. albicans* and *F. graminearum* with the following order of efficiency: ENN A = A1 > B1 > B = BEA and ENN A = B1 = BEA > A1 > B, respectively. Other filamentous fungi tested were found not sensitive to ENN or BEA with MIC values > 100 µM. Interestingly, the activity of ENN and BEA limited to *F. graminearum* may be useful to specifically target this important plant pathogen causing the fusarium head blight, a wheat disease associated to massive economic impact and to the presence of mycotoxins in food and feeds [36]. ENN and BEA were found to cause limited hemolysis of human erythrocytes with HC_50_ ranging from 68.7 to 534.5 µM with the following order of efficiency: ENN A > BEA > A1 > B1 > B. Taking into consideration the MIC of ENN A or B on *C. perfringens* (3.12 and 12.5 µM, respectively) and their HC_50_ (68.7 and 534.5 µM, respectively), this leads to interesting therapeutic index (TI, obtained by dividing the HC_50_ by the MIC values) of 22.0 and 42.7, respectively. Although poorly hemolytic, ENN and BEA were found to be highly toxic for nucleated human cells with IC_50_ on cell viability ranging from 0.003 to 54.2 µM, with ENN A and A1 being found the most toxic and BEA, ENN B, and B1 being found the less toxic in general. Taking into account the lower and the higher IC_50_ obtained respectively with ENN A1 on N87 cells and ENN B on HEK cells (0.003 and 54.2 µM, respectively) and their MIC on *C. perfringens* (3.12 and 12.5 µM, respectively), this leads to poor TI values ranging from 0.0009 to 4.3. Our results confirmed previous data showing that ENN and BEA are highly toxic to human cells, this toxicity being attributed to mitochondrial dysfunctions, ROS production, and cell death through apoptosis and/or necrosis [8]. Notably, fusafungine, a mixture of Enniatins, was once used for the treatment of upper respiratory tract infections [17]. This could be because safety tests were only performed previously using hemolysis assay and not cytotoxicity analysis. Fusafungine was then withdrawn from the market due to side effects, such side effects being explained by our study. In conclusion, this highlights the importance of safety analysis and scientists should always rely on different forms of toxicity analysis with different eukaryotic cells, not solely hemolysis.

Structurally, the cyclohexadepsipeptides ENN and BEA are very close, only differing at position R1, R2, and R3 (Figure 1). A structure–activity analysis could be performed for the antimicrobial and cytotoxic activities of these peptides (Table 5). Regarding the antibacterial activity of ENN and BEA, results show that globally the presence of *iso*-propyl group is detrimental to the antibacterial activity of these peptides. Thus, ENN B with three *iso*-propyl groups is the less active followed by ENN B1 having two *iso*-propyl and one *sec*-butyl groups, followed by ENN A1 with one *iso*-propyl and two *sec*-butyl groups and by ENN A with three *sec*-butyl groups. Accordingly, the presence and number of *iso*-propyl groups reduced the ability of the ENN to insert into bacterial lipids. Regarding BEA, the presence of the three phenylmethyl groups gave the same antibacterial activity than ENN A or A1 on *B. subtilis* and *S. aureus*. However, for *C. perfingens*, *E. faecalis,* and *M. smegmatis,* the presence of the phenylmethyl groups in BEA reduce the antibacterial activity to the same extent as for the presence of the *iso*-propyl groups in ENN B or B1, in accordance with its low ability to insert into bacterial lipids. Regarding the antifungal activity of ENN and BEA, the presence of *iso*-propyl and phenylmethyl groups was detrimental to the activity at least against *C. albicans*, ENN B, B1, and BEA being the less active on it. No clear conclusions can be drawn in the case of *F. graminearum* since ENN A, B1, and BEA showed higher activity compared to ENN A1 and B. Finally, regarding the hemolytic and cytotoxicity activities of ENN and BEA, the presence of *sec*-butyl and phenylmethyl groups in ENN A, A1, and BEA increases their insertion into POPC (i.e., the main lipid of the outer leaflet of the membrane of eukaryotic cells) and their hemolytic/toxic effect compared to ENN B and B1 containing *iso*-propyl groups. 

Although ENN and BEA were reported to possess antibacterial activity, the mechanism involved has not been previously described. Mechanistic approaches showed that ENN and BEA do not act by the same mechanism to kill bacteria. Although they are able to insert into bacterial lipids (at least ENN), neither ENN nor BEA possess pore-forming activity at MIC or 4 × MIC, only high doses (i.e., 100 µM) far away from their MIC (as low as 3.12 µM) are able to partially permeabilize the bacterial membrane. In accordance with their known ionophoric activity demonstrated on eukaryotic membranes and liposomes [17,27,28,29,30,31,32], our results showed that ENN and BEA are able to cause the depolarization of the bacterial membrane, potentially through the formation of ionic channels, at doses corresponding to their MIC. However, although this depolarization reached around 80% for ENN A and A1, it was partial (around 30–40% for ENN B, B1 and BEA). This result suggests that despite that ENN and BEA are able to act as ionophore on bacteria, this mechanism may not be the only one responsible for their antibacterial effect. Microscopic observation of the morphological changes caused by antibiotics, an alternative technique to radiolabeling developed by others and us to evaluate the effect of antibiotics on macromolecule synthesis [33,34], was thus used to further investigate the mechanism of action of ENN and BEA on bacteria. Results showed that in *C. perfringens*, ENN A and A1 (mainly containing *sec*-butyl groups) act like Rifampicin by causing an inhibition of the synthesis of RNA, whereas ENN B, B1, and BEA (mainly containing *iso*-propyl or phenylmethyl groups) act like Tetracycline by inhibiting the synthesis of proteins.

## 4. Conclusions

In this study, ENN and BEA were shown to exert an antimicrobial effect at relatively low concentrations. However, their use as a treatment for human bacterial infections is questionable due to their toxic nature. Importantly, results showed that the presence of either *iso*-propyl, *sec*-butyl, or phenylmethyl groups in these peptides can significantly influence their activities and mechanisms of action. 

## 5. Materials and Methods 

### 5.1. Chemicals

ENN A, A1, B, B1, and BEA were purchased from Sigma-Aldrich (Lyon, France) and have purity levels higher than 95–97%. Stock solutions of 1 mg/mL were prepared in absolute ethanol and stored at −20 °C. 

### 5.2. Microorganism Strains and Growth Conditions

Bacterial reference strains used in this study were obtained from either the American Type Culture Collection (ATCC), the German Leibniz Institute (DSMZ), or the French Pasteur Institute (CIP). Gram-negative bacterial strains used were: *Acinetobacter baumannii* (CIP 110431), *Escherichia coli* (ATCC 8739), *Helicobacter pylori* (ATCC 43504), and *Pseudomonas aeruginosa* (ATCC CRM-9027). Gram-positive bacterial strains used were: *Bacillus subtilis* (ATCC 6633), Nisin-resistant *B. subtilis* (DSMZ 347), *Clostridium perfringens* (ATCC 13124), Vancomycin-resistant *Enterococcus faecalis* (DSMZ 13591), *Staphylococcus aureus* (ATCC CRM-6538P), and Methicillin-resistant *S. aureus* strain MRSA USA300 (ATCC BAA-1717 USA 300 CA-MRSA). In addition to Gram-negative and Gram-positive strains, *Mycobacterium smegmatis* (mc2155, ATCC 700084) was also used as model of Mycobacterium species. All bacterial strains, except *C. perfringens, E. faecalis, H. pylori,* and *M. smegmatis*, were cultured on Luria Bertani (LB) agar plates and LB broth at 37 °C in aerobic condition. *E. faecalis* and *C. perfringens* were cultured in brain heart infusion (BHI) agar plates and BHI broth at 37 °C in an anaerobic chamber (Coy Laboratory Products, Grass Lake, MI, USA). *H. pylori* was grown in BHI agar plates and BHI broth at 37 °C in micro-aerobic condition using micro-aerobic BD GasPak generator. *M. smegmatis* was cultured in Middlebrook 7H10 agar plate and Middlebrook 7H9 broth at 37 °C in aerobic condition. Fungi used were *Aspergillus flavus* (DSMZ 1959), *A. niger* (ATCC 9142), *A. ochraceus* (DSMZ 824), *Candida albicans* (DSMZ 10697), *Fusarium graminearum* (DSMZ 1095), *F. oxysporum* (DSMZ 62316), *F. verticillioides* (DSMZ 62264), *Penicillium verrucosum* (DSMZ 12639), and *Stachybotrys chartarum* (DSMZ 2144). *C. albicans* was grown on LB agar plates and other fungi were grown on potatoes dextrose (PD) agar plates at prior to antifungal testing. 

### 5.3. Antimicrobial Activity Assay

Antimicrobial activity of ENN and BEA was evaluated through determination of the minimal inhibitory concentration (MIC) using two-fold serial dilutions of antimicrobial peptides in bacterial liquid media following the National Committee of Clinical Laboratory Standards (NCCLS, 1997) as previously described [1,37,38]. Briefly, single colonies of the different bacterial strains cultured on specific agar plates were used to inoculate 3 mL of Mueller–Hinton (MH) broth for all species except for *C. perfringens* and *M. smegmatis* that were cultured in 3 mL of BHI or Middlebrook 7H9 broth, respectively. Tubes were incubated overnight at 37 °C under agitation (200 rpm). The next day, bacteria solutions were diluted 1/100 in 3 mL of sterile medium and incubated at 37 °C, 200 rpm until bacteria reached log phase growth (optical density (OD) at 600 nm around 0.6). Bacteria were then diluted in sterile medium to reach bacterial density around 10^5^ bacteria/mL. Then, 100 μL of bacterial suspension were added per well to sterile polypropylene 96-well microplates (Greiner BioOne) and exposed to increasing concentrations of ENN or BEA (from 0 to 100 µM, 1:2 dilution). Volume of absolute ethanol corresponding to 100 µM of ENN or BEA was used as negative control and was found inactive. Plates were incubated at 37 °C for 18–24 h in aerobic (for *A. baumannii*, *E. coli*, *P. aeruginosa*, *B. subtilis*, *S. aureus*, *M. smegmatis*), anaerobic (for *C. perfingens* and *E. faecalis*), or micro-aerobic (for *H. pylori*) conditions except *M. smegmatis* that was incubated for 72 h in aerobic condition. The antifungal effect of ENN and BEA was tested following the reference methods for yeasts (NCCLS M27-A) and molds (M38-P) [39]. Liquid suspension of *C. albicans* was prepared by resuspending colonies collected from LB plates in sterile NaCl 0.85% solution. *C. albicans* were then diluted at 1–2 × 10^3^ yeasts/mL in RMPI media buffered with MOPS (final concentration of 0.165 mol/L (pH 7.0)). For filamentous fungi, conidi were collected from fungi grown on PDA plates using sterile solution of NaCl 0.9% supplemented with Tween at 0.1%. After counting under microscope, dilution at 2–3 × 10^4^ conidi/mL were prepared in RMPI media buffered with MOPS (final concentration of 0.165 mol/L (pH 7.0)). Diluted yeast or fungi were then subjected to MIC testing as described for bacteria. For *C. albicans*, plates were incubated at 35 °C for 24 h before reading. For other fungi, plates were incubated for 3 to 4 days before reading. At the end of the incubation, OD_600nm_ was measured using microplate reader (Biotek, Synergy Mx, Colmar, France). The MIC was defined as the lowest concentration of drug that completely inhibited visible growth of the organism. Experiments were conducted in triplicate (*n* = 3). 

### 5.4. Hemolytic Activity Assay

The hemolytic activity of the peptides was determined by the leakage of hemoglobin from human erythrocytes (obtained from Divbioscience, Ulvenhout, Netherlands) as previously described [1,38,40]. Briefly, human erythrocytes were washed 2 times with sterile phosphate buffer saline (PBS, pH 7.4) (from Thermofisher, Les Ulis, France) and pelleted using centrifugation at 800× *g* for 5 min. Human erythrocytes were then resuspended in PBS at a final concentration of 8%. Then, 150 μL of human erythrocytes were added per well of sterile 96-well microplates and exposed to increasing concentrations of ENN or BEA (from 0 to 100 µM, 1:2 dilution). Volume of absolute ethanol corresponding to 100 µM of ENN or BEA was used as negative control and was found inactive. After 1 h at 37 °C, the microplates were centrifuged at 800× *g* for 5 min. Then, 100 μL of supernatant were carefully collected and transferred to new 96-well microplates and OD_405nm_ was measured using microplate reader (Biotek, Synergy Mx, Colmar, France). Hemolysis caused by ENN or BEA was expressed as percentage of hemolysis, Triton-X100 at 0.1% (*v*:*v*) being used as positive control giving 100% hemolysis. The HC_50_ values of ENN and BEA (i.e., the concentrations causing either 50% of hemolysis) were calculated using GraphPad^®^ Prism 7 software (San Diego, CA, USA). Experiments were conducted in triplicate (*n* = 3).

### 5.5. Cytotoxic Assays

The impact of ENN or BEA on the viability of human cells were evaluated as previously described [37,38,41]. Cells used were human normal lung epithelial cells BEAS-2B (ATCC CRL-9609), human intestinal cell line Caco-2 (ATCC HTB-37), human normal epidermal keratinocytes (HEK) (from Lonza), human liver cell line HEPG2 (ATCC HB-8065), human normal endothelial cells (Human Umbilical Vein Endothelial Cells, HUVEC) (from Sigma-Aldrich, Lyon, France), and human gastric cell line N87 (ATCC CRL-5822). BEAS-2B, Caco-2 and HEPG2 cells were cultured in DMEM supplemented with 10% fetal calf serum (FCS), 1% l-glutamine, and 1% antibiotics (all from Invitrogen (Carlsbad, CA, USA). N87 cells were cultured in RPMI media supplemented with 10% fetal calf serum (FCS), 1% L-glutamine, and 1% antibiotics (all from Invitrogen, Carlsbad, CA, USA). HEK and HUVEC cells were cultured in specific medium (from Lonza and Sigma Aldrich, respectively). Cells were routinely grown on 25 cm^2^ flasks and maintained in a 5% CO_2_ incubator at 37 °C. For toxicity assay, human cells grown on 25 cm^2^ flasks were detached using trypsin-EDTA solution (from Thermofisher), counted using Malassez counting chamber, diluted in appropriate culture media, and seeded into 96-well cell culture plates (Greiner bio-one, Paris, France) at approximately 10^4^ cells per well. The cells were left to grow for 48–72 h at 37 °C in a 5% CO_2_ incubator until they reached confluence. Media from wells was then aspirated and cells were treated with 100 µL of appropriate culture media containing increasing concentrations of ENN or BEA (from 0 to 100 µM, 1:2 serial dilution). Volume of absolute ethanol corresponding to 100 µM of ENN or BEA was used as negative control and was found inactive. The plates were then incubated at 37 °C for 48 h. Resazurin-based in vitro toxicity assay kit (from Sigma-Aldrich, Lyon, France) was then used to assess the viability of the cells following manufacturer’s instructions. Briefly, resazurin stock solution was diluted 1:10 in sterile PBS containing calcium and magnesium (PBS^++^, pH 7.4). Plates were aspirated and 100 µL of the diluted solution were added per wells. After 4 h incubation at 37 °C, fluorescence intensity was measured using microplate reader (Biotek, Synergy Mx, Colmar, France) with an excitation wavelength of 530 nm and an emission wavelength of 590 nm. The fluorescence values were normalized by the controls (untreated cells) and expressed as percentage of cell viability. The IC_50_ values of ENN or BEA (i.e., the concentrations causing a reduction of 50% of the cell viability) were calculated using GraphPad^®^ Prism 7 software (San Diego, CA, USA). Experiments were conducted in triplicate (*n* = 3).

### 5.6. Peptid–eLipid Interaction Assay

Peptide–lipid interaction was evaluated using reconstituted lipid monolayer at the air–water interface as previously described [38,42,43]. Pure prokaryotic and eukaryotic lipids used were: POPC (1-palmitoyl-2-oleoyl-glycero-3-phosphocholine), POPE (1-palmitoyl-2-oleoyl-sn-glycero-3-phosphoethanolamine), POPG (1-palmitoyl-2-oleoyl-sn-glycero-3-phospho-(1’-rac-glycerol), and cardiolipin, all from Avanti Polar Lipid (Alabama, USA). LTA (Lipoteichoic acid from *B. subtilis*) was obtained from Invivogen SAS (Toulouse, France). Pure lipids were reconstituted in chloroform:methanol (2:1, *v*:*v*) at 1 mg/mL and stored at −20  °C under nitrogen. The interaction and insertion of the ENN and BEA into lipids were evaluated through the measurement of their critical pressure of insertion as previously described [38,43]. Briefly, lipids were spread using a 50 µL Hamilton’s syringe at the surface of 800 µL of sterile PBS creating a lipid monolayer at the air–water interface. Lipids were added until the surface pressure reached the desired value. After 5–10 min of incubation allowing evaporation of the solvent and stabilization of the initial surface pressure, 8 µL of ENN or BEA diluted in sterile PBS at 100 µM were injected into the 800 µL sub-phase of PBS under the lipid monolayer (pH 7.4, volume 800 µL) using a 10 µL Hamilton syringe giving a final concentration of peptides of 1 µM, preliminary experiments having shown that this concentration gave optimal insertion. The variation of the surface pressure caused by peptide injection was then continuously monitored using a fully automated microtensiometer (µTROUGH SX, Kibron Inc., Helsinki, Finland) until reaching equilibrium (maximal surface pressure increase being usually obtained after 15–25 min). The critical pressure of insertion of each peptide was determined by changing the initial pressure of lipid monolayer (from 10 and 30 mN/m) and measuring the variation of pressure caused by the insertion of the peptide. The critical pressure of insertion of each peptide in a particular lipid was determined graphically and calculated as the theoretical value of initial pressure of lipid monolayer not permissive to peptide insertion, i.e., causing a variation of pressure equal to 0 mN/m. 

### 5.7. Bacterial Membrane Permeabilization Assay

Membrane permeabilization by ENN or BEA was evaluated using the cell-impermeable DNA/RNA probe propidium iodide as previously explained [38,42]. Logarithmic growing bacterial suspension of *C. perfringens* was prepared from over-night bacterial suspension by 1 in 10 dilution. After 3 h incubation at 37 °C, 200 rpm, bacterial suspensions were centrifuged for 5 min at 3000× *g*. Cell pellets were then resuspended in sterile PBS at about 10^9^ bacteria/mL. Propidium iodide (from Sigma Aldrich, Lyon, France) was then added to the suspension at a final concentration of 60 µM. Then, 100 µL of this suspension were transferred into 96-well black plates and exposed to increasing concentrations of ENN or BEA (from 0 to 100 µM, 1:2 dilution). Volume of absolute ethanol corresponding to 100 µM of ENN or BEA was used as negative control and was found inactive. All steps of the assay were performed in anaerobic condition using an anaerobic chamber (Coy Laboratory Products, Grass Lake, MI). Bacteria were incubated at 37 °C and the kinetics of fluorescence variations (excitation at 530 nm and emission at 590 nm) were then recorded over time using a microplate reader (Biotek, Synergy Mx, Colmar, France). Cetyl trimethylammonium bromide (CTAB) at 300 µM was used as positive control giving 100% permeabilization. Results were expressed in % of permeabilization. All experiments were done in triplicate.

### 5.8. Membrane Depolarization Assay

Membrane depolarization caused by ENN or BEA was evaluated using the potentiometric probe 3,3’-Dipropylthiadicarbocyanine Iodide (DiSC_3_(5)) as previously explained [44] with some changes. Logarithmic growing bacterial suspension of *C. perfringens* was prepared from overnight bacterial suspension by 1 in 10 dilution. After 3 h of incubation at 37 °C (OD_600_ around 1), DiSC_3_(5) (Sigma-Aldrich, Lyon, France) was added to the suspension at a final concentration of 2 µM. Then, 100 µL of this suspension were transferred into 96-well black plates in anaerobic conditions. Plates were then incubated for around 3 h until fluorescence signal (excitation at 622 nm and emission at 670 nm) reaches minimum indicating complete quenching of the probe in the bacterial membrane. Bacteria were then exposed to increasing concentrations of ENN or BEA (from 0 to 100 µM, 1:2 dilution). Kinetics of fluorescence variations (excitation at 622 nm and emission at 670 nm) were then recorded over time at 37 °C in anaerobic conditions using a microplate reader (Biotek, Synergy Mx, Colmar, France). Volume of absolute ethanol corresponding to 100 µM of ENN or BEA was used as negative control and was found inactive. Preliminary experiments showed that low concentration of CTAB (i.e., 3 µM) is able to cause membrane depolarization in the absence of permeabilization and was thus used as positive control giving 100% depolarization. Results were expressed in % of depolarization. All experiments were done in triplicate.

### 5.9. Determination of the Mechanism of Action on Bacteria Using Microscopic Observation

Microscopic observation was used to determine the mechanism of action of ENN and BEA on bacteria as previously described [33,34]. Briefly, overnight culture of *C. perfringens* was diluted 1:100 in BHI broth and grown at 37 °C in anaerobic condition until it reached an OD_600nm_ of about 0.2. Conventional antibiotics, ENN, or BEA were then added at a concentration equivalent to 5 times their MIC and bacteria were incubated at 37 °C for 4 h in anaerobic conditions. At the end of the incubation, bacteria were stained for 15 min on ice for membrane and DNA using 12 μg/mL FM4-64FX (from Thermofisher, Les Ulis, France) and 2 μg/mL DAPI (from Sigma-Aldrich, Lyon, France), respectively. After staining, bacteria were pelleted by centrifugation (centrifugation for 30 s at 7500× *g*) and resuspended in sterile PBS. Bacteria were pelleted a second time (centrifugation for 30 s at 7500× *g*) and resuspended in 500 µL of sterile PBS. Bacteria were then pelleted a last time and fixed with PBS plus paraformaldehyde (PFA) at 4% (*v*:*v*) for 15 min on ice in the dark. After centrifugation for 1 min at 7500× *g*, the samples were resuspended in 50 µL of mounting solution (Vector Laboratories, Peterborough, PE2 6XS, UK). Then, 8 µL were transferred onto microscope glass slides and bacteria were observed and images collected using an IX71 Fluoview confocal microscope (Olympus, Rungis, France) with excitation at 543 nm and 405 nm for FM4-64FX and DAPI, respectively. Positive controls used were the following conventional antibiotics: Amoxicillin, Gemifloxacin, Rifampicin, and Tetracycline used at 5 times their MIC (i.e., 14.5 µM, 1.25 µM, 0.95 µM, and 0.146 µM, respectively).

## Figures and Tables

**Figure 1 toxins-11-00514-f001:**
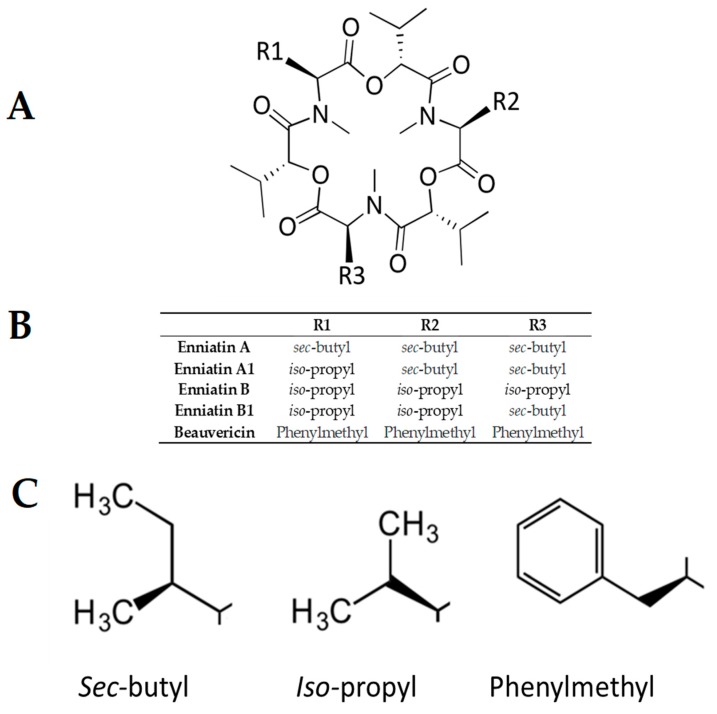
Structure of enniatin (ENN) and beauvericin (BEA). (**A**) General structure of the cyclohexadepsipeptides scaffold, (**B**) chemical nature of the side chains R1, R2, and R3, (**C**) chemical structure of the side chains present in ENN and BEA.

**Figure 2 toxins-11-00514-f002:**
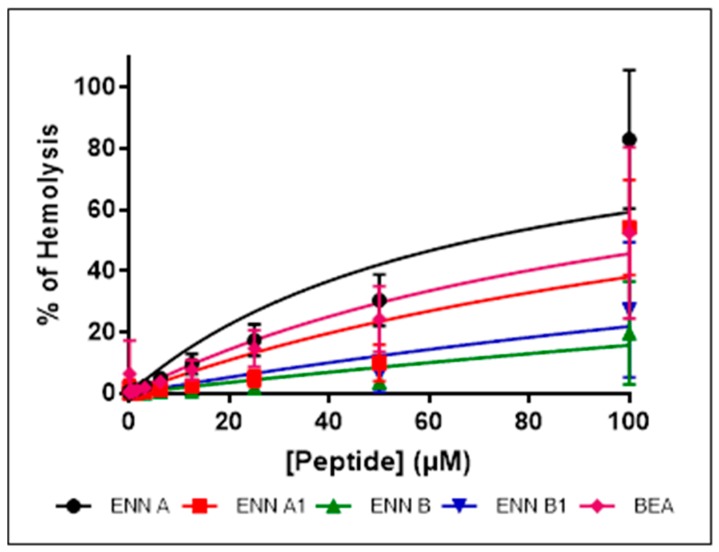
Hemolytic activity of ENN and BEA on human erythrocytes. Hemolysis of human red blood cells was measured after exposure to increasing concentrations of ENN or BEA. Results are expressed as percentage of hemolysis, Triton X-100 being used as positive control giving 100% permeabilization. Results are expressed as means ± SD (*n* = 3).

**Figure 3 toxins-11-00514-f003:**
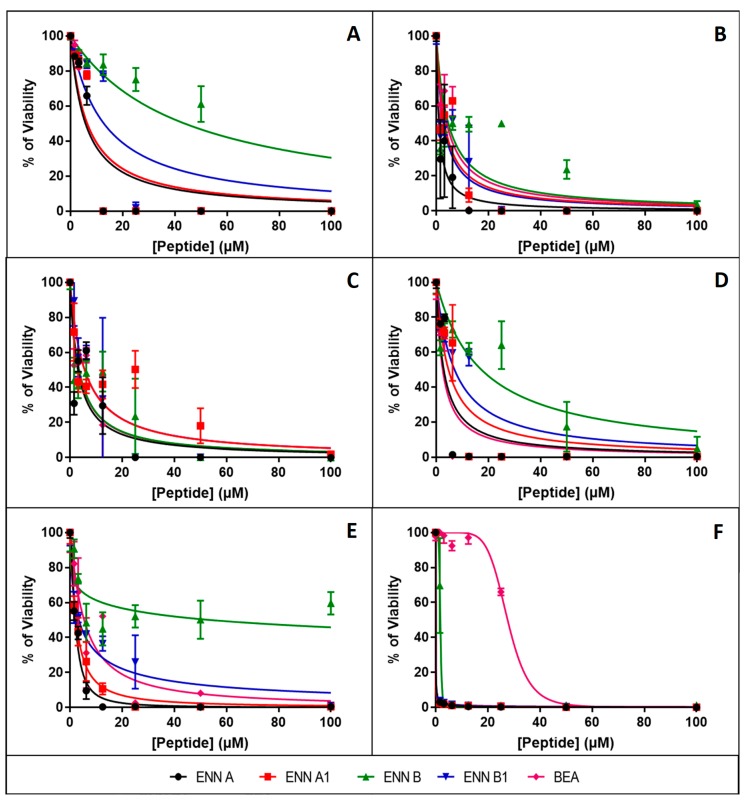
Cytotoxicity of ENN and BEA against human nucleated cells. Toxicity induced by 48 h exposure to increasing concentrations of ENN or BEA was measured using human cells BEAS-2B (**A**), Caco-2 (**B**), HEPG2 (**C**), HUVEC (**D**), HEK (**E**), and N87 (**F**). Results are expressed as percentage of cell viability (means ± SD (*n* = 3)).

**Figure 4 toxins-11-00514-f004:**
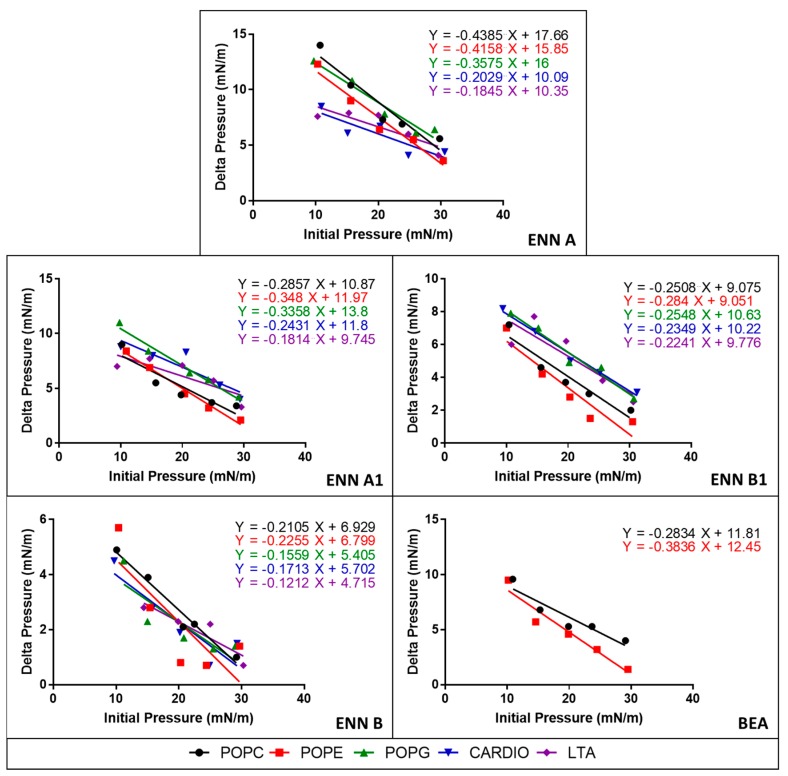
Evaluation of the insertion of ENN and BEA into eukaryotic and bacterial lipids. The insertion of ENN and BEA into prokaryotic or eukaryotic pure lipids was measured using lipid monolayers. Data are plotted as delta pressure (difference between final and initial pressure) with respect to initial pressure at injection of peptides.

**Figure 5 toxins-11-00514-f005:**
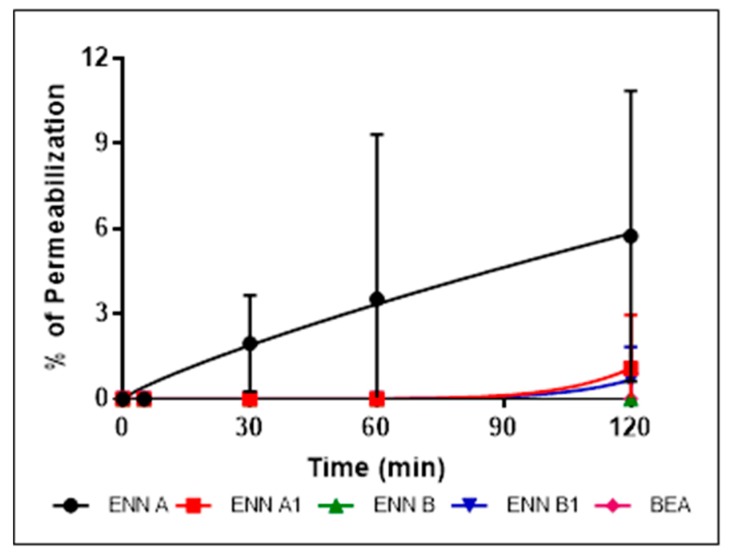
Effect of ENN and BEA at their MIC on the membrane integrity of *C. perfringens*. *C. perfringens* were exposed to ENN or BEA at concentrations corresponding to their MIC values. The effect of the peptides on the integrity of the bacterial membrane was followed over 120 min using propidium iodide. Percentage of permeabilization caused by ENN or BEA was calculated using the detergent cetyl trimethylammonium bromide (CTAB) at 300 µM as positive control giving 100% effect. Results are expressed as means ± SD (*n* = 3).

**Figure 6 toxins-11-00514-f006:**
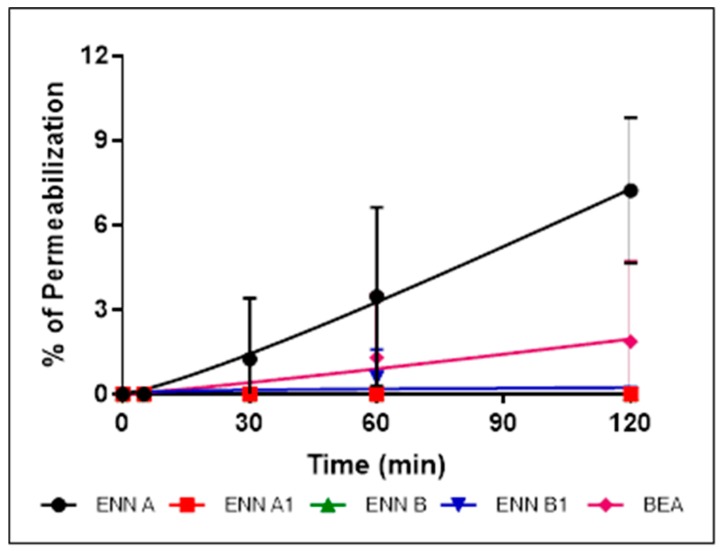
Effect of ENN and BEA at four times their MIC on the membrane integrity of *C. perfringens*. *C. perfringens* were exposed to ENN or BEA at concentrations corresponding to four times their MIC values. The effect of the peptides on the integrity of the bacterial membrane was followed over 120 min using propidium iodide. Percentage of permeabilization caused by ENN or BEA was calculated using the detergent CTAB at 300 µM as positive control giving 100% effect. Results are expressed as means ± SD (*n* = 3).

**Figure 7 toxins-11-00514-f007:**
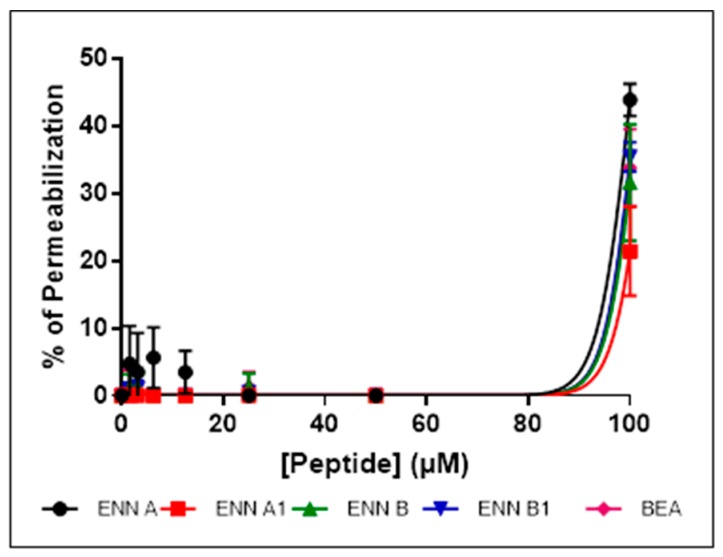
*C. perfringens* were exposed to increasing concentrations of ENN or BEA (from 0 to 100 µM) for 120 min at 37 °C. The effect of the peptides on the integrity of the bacterial membrane was then measured using propidium iodide. Percentage of permeabilization caused by ENN or BEA was calculated using the detergent CTAB at 300 µM as positive control giving 100% effect. Results are expressed as means ± SD (*n* = 3).

**Figure 8 toxins-11-00514-f008:**
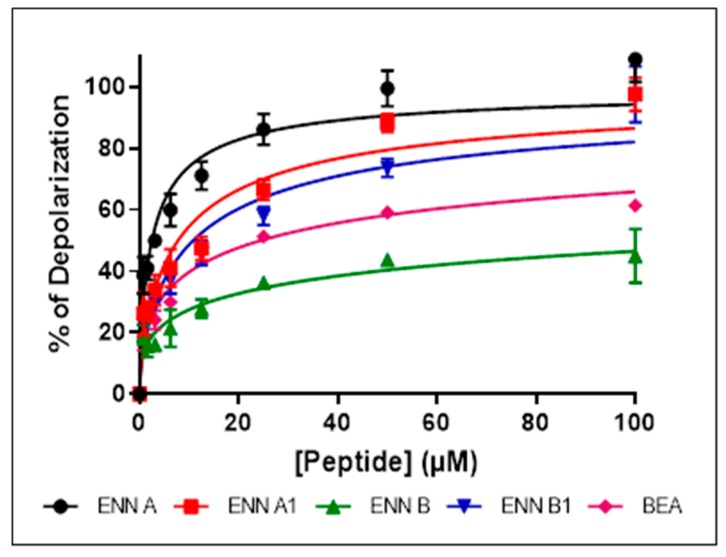
Dose-dependent effect of ENN and BEA on the membrane potential of *C. perfringens*. Dose-dependent effect of ENN or BEA on the membrane potential of *C. perfringens* was measured using the potentiometric probe DiSC3(5). The detergent CTAB was used at 3 µM as positive control giving 100% depolarization. Results are expressed as means ± SD (*n* = 3).

**Figure 9 toxins-11-00514-f009:**
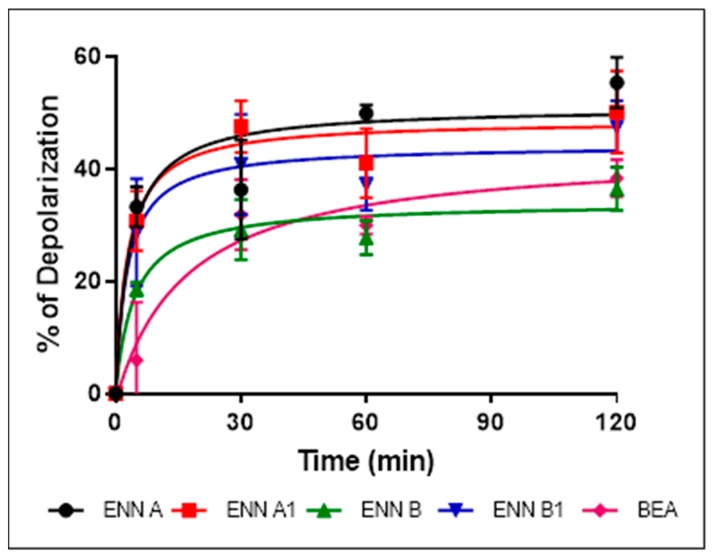
Effect of ENN and BEA at their MIC on the membrane potential of *C. perfringens*. The time-dependent effect of ENN or BEA at their MIC on the membrane potential of *C. perfringens* was measured using the potentiometric probe DiSC3(5). The detergent CTAB was used at 3 µM as positive control giving 100% depolarization. Results are expressed as means ± SD (*n* = 3).

**Figure 10 toxins-11-00514-f010:**
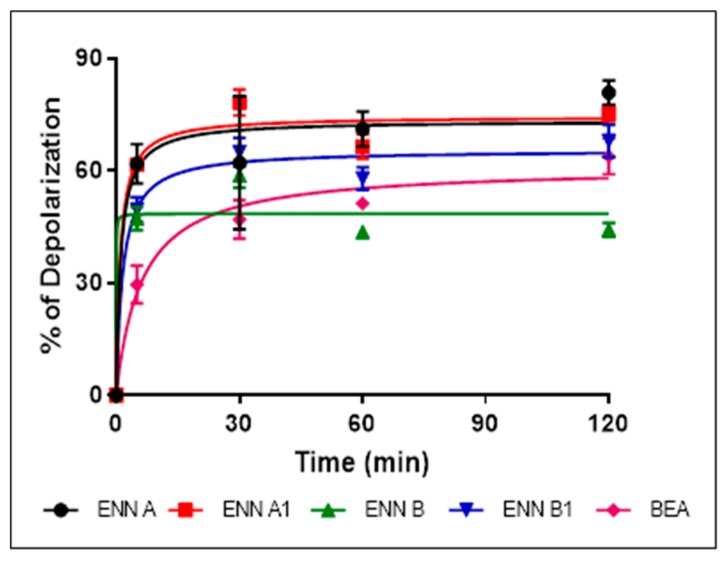
Effect of ENN and BEA at four times their MIC on the membrane potential of *C. perfringens*. The time-dependent effect of ENN or BEA at four times their MIC on the membrane potential of *C. perfringens* was measured using the potentiometric probe DiSC3(5). The detergent CTAB was used at 3 µM as positive control giving 100% depolarization. Results are expressed as means ± SD (*n* = 3).

**Figure 11 toxins-11-00514-f011:**
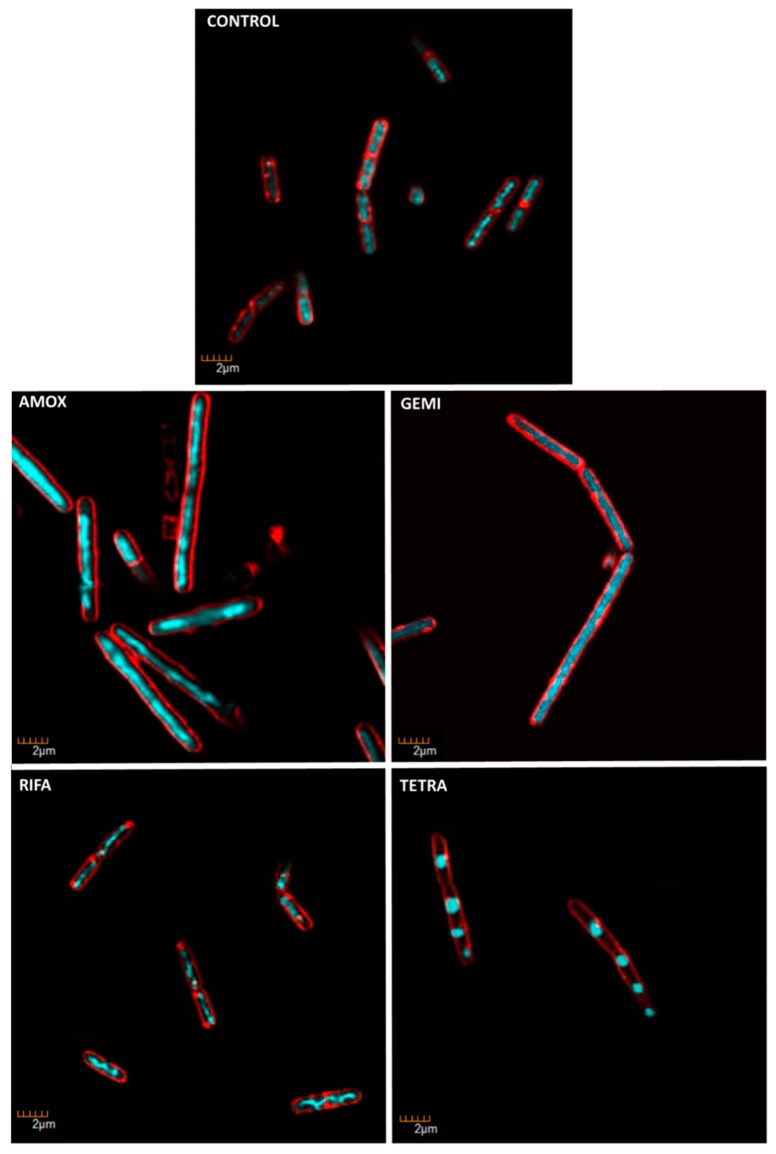
*C. perfringens* bacteria were incubated for 4 h at 37 °C in anaerobic conditions with 5 × MIC of conventional antibiotics with known bacterial target. Antibiotic used were: Amoxicillin (AMOX, inhibiting cell wall synthesis), Gemifloxacin (GEMI, inhibiting DNA synthesis), Rifampicin (RIFA, inhibiting RNA synthesis), and Tetracycline (TETRA, inhibiting protein synthesis). At the end of the incubation, bacteria were fixed and membrane and DNA were stained using FM4-64FX (red signal) and DAPI (blue signal), respectively. Bacteria were observed using confocal microscope. Images representative of observed effects in triplicate are shown.

**Figure 12 toxins-11-00514-f012:**
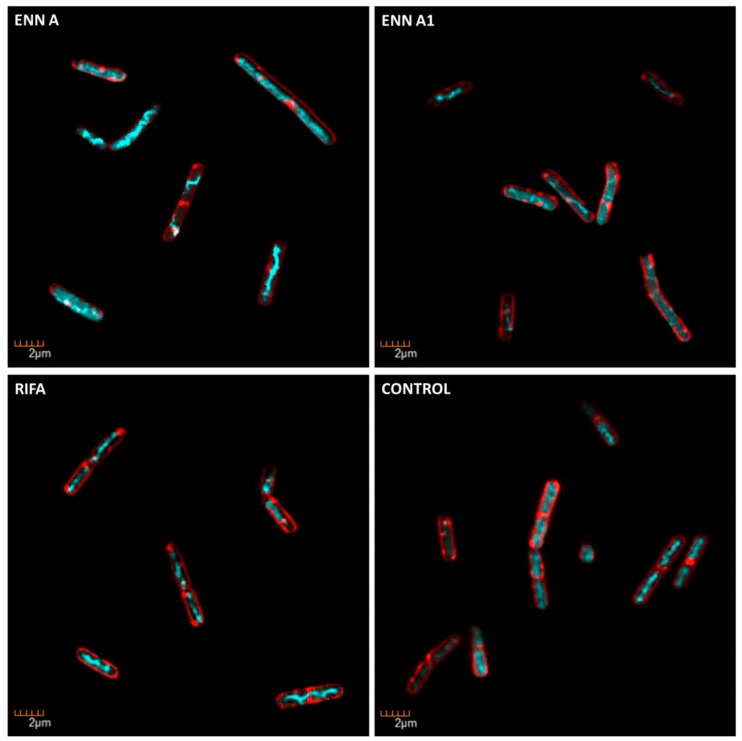
Morphological changes caused by ENN A and A1 in *C. perfringens*. *C. perfringens* bacteria were incubated for 4 h at 37 °C in anaerobic conditions with 5 X MIC of ENN A or A1. At the end of incubation, bacteria were fixed and membrane and DNA were stained using FM4-64FX (red signal) and DAPI (blue signal), respectively. Bacteria were observed using confocal microscope. Images representative of observed effects in triplicate are shown. The effect of Rifampicin (RIFA, inhibiting RNA synthesis) is shown for comparison.

**Figure 13 toxins-11-00514-f013:**
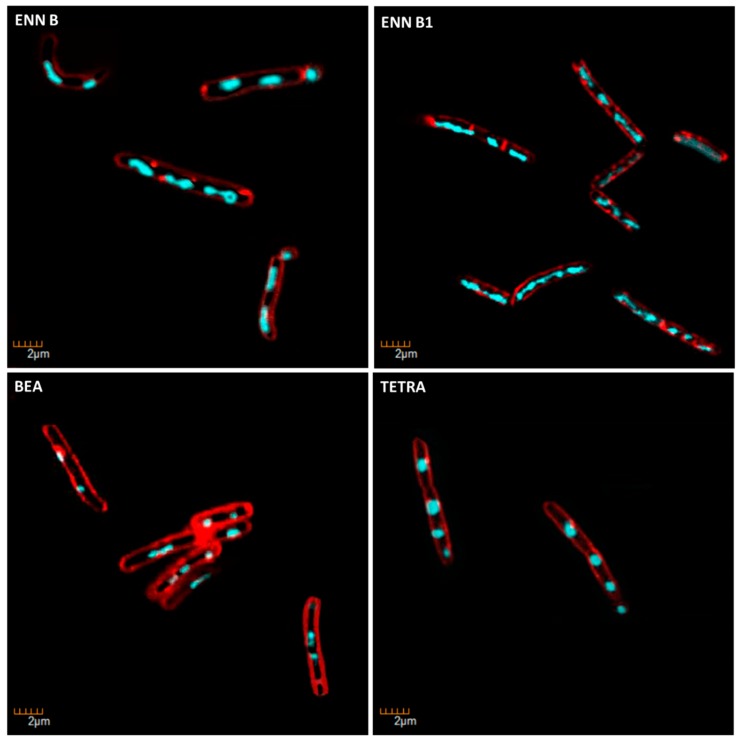
Morphological changes caused by ENN B, B1, and BEA in *C. perfringens*. *C. perfringens* bacteria were incubated for 4 h at 37 °C in anaerobic conditions with 5 × MIC of ENN B, B1, or BEA. At the end of incubation, bacteria were fixed and membrane and DNA were stained using FM4-64FX (red signal) and DAPI (blue signal), respectively. Bacteria were observed using confocal microscope. Images representative of observed effects in triplicate are shown. The effect of Tetracycline (TETRA, inhibiting protein synthesis) is shown for comparison.

**Table 1 toxins-11-00514-t001:** Minimum inhibitory concentration (MIC) values of ENN and BEA on microorganisms. MIC of ENN and BEA on various microorganisms were determined three times and are expressed in micromolar (µM).

Microorganisms		ENN A	ENN A1	ENN B	ENN B1	BEA
Gram positive bacteria	*B. subtilis*	6.25–12.5	12.5–25	>100	25–50	6.25–12.5
	*B. subtilis* NR	6.25–12.5	25	>100	50	6.25–12.5
	*C. perfringens*	3.12	3.12–6.25	12.5	6.25	6.25
	*E. faecalis*	3.12	6.25	>100	12.5	12.5
	*S. aureus*	6.25–12.5	12.5–25	>100	25	12.5
	*S. aureus* MRSA	6.25–12.5	12.5–25	>100	25	6.25–12.5
Gram negative bacteria	*A. baumannii*	>100	>100	>100	>100	>100
	*E. coli*	>100	>100	>100	>100	>100
	*H. pylori*	>100	>100	>100	>100	>100
	*P. aeruginosa*	>100	>100	>100	>100	>100
Mycobacteria	*M. smegmatis*	6.25	3.12	100	6.25	25
Fungi	*A. flavus*	>100	>100	>100	>100	>100
	*A. niger*	>100	>100	>100	>100	>100
	*A. ochraceus*	>100	>100	>100	>100	>100
	*C. albicans*	1.5	1.5	6.25	3.12	6.25
	*F. graminearum*	25	50	>100	25	25
	*F. oxysporum*	>100	>100	>100	>100	>100
	*F. verticillioides*	>100	>100	>100	>100	>100
	*P. verrucosum*	>100	>100	>100	>100	>100
	*S. chartarum*	>100	>100	>100	>100	>100

*C. perfringens* was found to be the most sensitive strain to all peptides, including ENN B, with MIC values ranging from 3.12 to 12.5 µM and the following order of efficiency: ENN A = A1 > B1 = BEA > B. For *B. subtilis* and *S. aureus*, MIC values ranged from 6.25 to >100 µM with the following order of efficiency: ENN A = BEA > A1 > B1 > B. Finally, for *E. faecalis*, MIC values ranged from 3.12 to > 100 µM with the following order: ENN A > A1 > B1 = BEA > B. Importantly, in the context of the worldwide increase in antibiotic resistance, ENN and BEA were found to be active against Gram-positive bacteria resistant to antibiotics such as Nisin-resistant *B. subtilis,* Vancomycin-resistant *E. faecalis,* and Methicillin-resistant *S. aureus*. ENN and BEA were active against *Mycobacterium*; *M. smegmatis* gave MIC values ranging from 3.12 to 100 µM in the following order: ENN A1 > A = B1 > BEA >> B. Regarding antifungal activity, peptides were found to be active against *C. albicans* with MIC values ranging from 1.5 to 12.5 µM and the following order of efficiency: ENN A = A1 > B1 > B = BEA. Conversely, the peptides showed no activity against filamentous fungi, except *F. graminearum* that was sensitive with MIC values ranging from 25 to > 100 µM with the following order: ENN A = B1 = BEA > A1 > B.

**Table 2 toxins-11-00514-t002:** Effect of ENN and BEA on different human cells. The HC50 (i.e., the dose causing 50% hemolysis) and the IC_50_ on cell viability of human nucleated cells (i.e., the dose causing 50% reduction in the cell viability) were determined from Figure 2; Figure 3, respectively. Values are expressed in micromolar (µM) (means ± S.D, *n* = 3).

Cells	ENN A	ENN A1	ENN B	ENN B1	BEA
**BEAS-2B**	5.7 ± 1.2	6.4 ± 1.5	43.7 ± 7.7	12.7 ± 2.8	6.3 ± 1.4
**Caco-2**	1.1 ± 0.2	2.7 ± 0.5	4.6 ± 1.3	3.1 ± 0.5	3.9 ± 0.7
**HEK**	2.0 ±0.1	2.3 ± 0.1	54.2 ± 35.3	3.3 ± 0.5	5.4 ± 0.9
**HEPG2**	3.0 ± 0.7	5.6 ± 1.1	3.4 ± 0.7	5.6 ± 1.4	3.4 ± 0.6
**HUVEC**	2.8 ± 0.6	4.6 ± 0.9	17.3 ± 3.3	7.0 ± 1.1	2.4 ± 0.5
**N87**	0.01 ± 0.01	0.003 ± 0.002	1.7 ± 0.1	0.008 ± 0.008	27.5 ± 0.7
**Erythrocytes**	68.7 ± 9.6	162.7 ± 22.8	534.5 ± 86.5	356.4 ± 59.3	118.9 ± 15.8

**Table 3 toxins-11-00514-t003:** Evaluation of the insertion of ENN and BEA into eukaryotic and bacterial lipids. The critical pressure of insertion (expressed in mN/m) of ENN and BEA into pure lipids present in the outer membrane leaflet of the bacterial (phosphatidyl-ethanolamine (POPE), phosphatidyl-glycerol (POPG), Cardiolipin, lipoteichoic acid (LTA)) or of the eukaryotic membrane (phosphatidyl-choline, POPC) was calculated from the equations shown in Figure 4. -- indicates the absence of interaction.

*Peptides*	POPC	POPE	POPG	Cardiolipin	LTA
**ENN A**	40.28	38.12	44.75	49.72	56.1
**ENN A1**	38.06	34.38	41.10	48.57	53.71
**ENN B**	32.92	30.15	34.67	33.29	38.9
**ENN B1**	36.19	31.87	41.71	43.51	43.62
**BEA**	41.65	32.46	--	--	--

**Table 4 toxins-11-00514-t004:** Description of the morphological changes caused by conventional antibiotics, ENN or BEA in *C. perfringens*. Morphological changes of *C. perfringens* caused by conventional antibiotics, ENN, or BEA illustrated in Figure 11, Figure 12 and Figure 13 are summarized in this table.

Condition	Known/possible target	Observed Morphology
**Control**	-	No cell elongation Normally condensed DNA
**Amoxicillin**	Cell wall biosynthesis	Elongated cells (5–6 times compared to control) Normally condensed DNA
**Gemifloxacin**	DNA synthesis	Elongated cells (5–6 times compared to control) Decondensed DNA Chain formation
**Rifampicin**	RNA biosynthesis	No cell elongation DNA condensed in one filament crossing the cells
**Tetracycline**	Protein biosynthesis	Some cell elongation (1.5–2 times compared to control) DNA condensed in 1–4 circles per cells
**ENN A**	RNA biosynthesis	Some cell elongation DNA condensed in one filament crossing the cells
**ENN A1**	RNA biosynthesis	No cell elongation DNA condensed in one filament crossing the cells
**ENN B**	Protein biosynthesis	Some cell elongation (1.5–2 times compared to control) DNA condensed in 1–4 circles per cells
**ENN B1**	Protein biosynthesis	Some cell elongation (1.5–2 times compared to control) DNA condensed in 1–4 circles per cells
**BEA**	Protein biosynthesis	No cell elongation DNA condensed in 1–4 circles per cells

**Table 5 toxins-11-00514-t005:** Structure–activity analysis. This table summarizes the consequences of the presence of *iso*-propyl, *sec*-butyl, or phenylmethyl groups at positions R1, R2, or R3 in ENN and BEA on their activities.

Chemical Groups Present at R1, R2, R3	Effects
***iso*-propyl**	• Decrease in antimicrobial activity
	• Decrease in hemolysis
	• Variable effect on cytotoxicity (either a decrease for BEAS-2B, Caco-2, HEK, and HUVEC cells or an increase for HEPG2 and N87 cells)
	• Decrease in insertion in bacterial and eukaryotic lipids
	• Inhibition of bacterial protein synthesis
***sec*-butyl**	• Increase in antimicrobial activity
	• Increase in hemolysis
	• Increase in cytotoxicity
	• Increase in insertion in bacterial and eukaryotic lipids
	• Inhibition of bacterial RNA synthesis
**Phenylmethyl**	• Variable effect on antimicrobial activity (either a decrease for *C. albicans, C. perfringens, E. faecalis,* and *M. smegmatis* or an increase for *B. subtilis and S. aureus*)
	• Increase in hemolysis
	• Variable effect on cytotoxicity (either a decrease for BEAS-2B, Caco-2, HEK, HEPG2, and N87 cells or an increase for HUVEC cells)
	• Increase in insertion in eukaryotic lipid (POPC)
	• Inhibition of bacterial protein synthesis

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
