# Peer review of "Comparative Structure–Activity Analysis of the Antimicrobial Activity, Cytotoxicity, and Mechanism of Action of the Fungal Cyclohexadepsipeptides Enniatins and Beauvericin"

_toxins, 2019, doi:10.3390/toxins11090514_

Round 1
Reviewer 1 Report
The manuscript entitled “Comparative structure-activity analysis of the 2 antimicrobial activity, cytotoxicity and mechanism of 3 action of the fungal cyclohexadepsipeptides 4 Enniatins and Beauvericin” is a thorough study of promising new types of potential antibiotics from the family of as antimicrobial peptides. It shows traditional experiments such as antibacterial and antifungal properties as well as hemolytic toxicity but also shows cytotoxicity of eukaryotic cells and certainly less common experiments as membrane integrity, permeabilization, depolarization and morphological changes in cell membrane. It is certainly a thorough study of these cyclopeptides that will help and inspire readers for their work in the field of antimicrobial peptides.
Said that I would suggest only few minor things:
- For every figure, the author repeats that the results were measured or calculated or studied “as described in the Materials and Methods section” on the captions under the figure. I would suggest to mention that only once or not at all in the captions. Maybe once in the text.
- Also in the same figure captions for plots the author shows the icon and colors for each peptide described on the plots but they are also mentioned every time in the captions under the figures. Eg: “ENN A is in black circle, ENN A1 231 in red square, ENN B in green triangle, ENN B1 in blue inverted triangle, and BEA in violet diamond”. I would suggest to eliminate one or the other.
Aside of that, I do not find any other issue for this manuscript. This is one of those works where you not only get new data and learn something new but also enjoy reading. I would highly recommend publishing it.
Author Response
Dear Editor, Dear Reviewer,
First we would like to thank you for giving us the opportunity to submit a revised version of our work. We would like also to thank you for your helpful comments and suggestions that helped us to improve the quality of our work. We hope our answers will satisfy the reviewer. Please find below the list of questions and our answers to them. regards
Reviewer 1’s comments:
Comment 1 : The manuscript entitled “Comparative structure-activity analysis of the 2 antimicrobial activity, cytotoxicity and mechanism of 3 action of the fungal cyclohexadepsipeptides 4 Enniatins and Beauvericin” is a thorough study of promising new types of potential antibiotics from the family of as antimicrobial peptides. It shows traditional experiments such as antibacterial and antifungal properties as well as hemolytic toxicity but also shows cytotoxicity of eukaryotic cells and certainly less common experiments as membrane integrity, permeabilization, depolarization and morphological changes in cell membrane. It is certainly a thorough study of these cyclopeptides that will help and inspire readers for their work in the field of antimicrobial peptides.
Answer to comment 1 : We would like to thank reviewer 1 for her/his very nice comment about our work.
Comment 2 : For every figure, the author repeats that the results were measured or calculated or studied “as described in the Materials and Methods section” on the captions under the figure. I would suggest to mention that only once or not at all in the captions. Maybe once in the text.
Answer to comment 2 : Accordingly to reviewer’s suggestion, we removed this mention in all captions of the revised manuscript.
Comment 3 : Also in the same figure captions for plots the author shows the icon and colors for each peptide described on the plots but they are also mentioned every time in the captions under the figures. Eg: “ENN A is in black circle, ENN A1 231 in red square, ENN B in green triangle, ENN B1 in blue inverted triangle, and BEA in violet diamond”. I would suggest to eliminate one or the other.
Answer to comment 3 : Accordingly to reviewer’s suggestion, we removed this mention in all captions of the revised manuscript.
Comment 4 : Aside of that, I do not find any other issue for this manuscript. This is one of those works where you not only get new data and learn something new but also enjoy reading. I would highly recommend publishing it.
Answer to comment 4 : We would to thank the reviewer for this very nice comment.
Reviewer 2 Report
The manuscript reports the biological characterization of five fungal cyclohexadepsipeptides. In particular their evaluated the antimicrobial activity and cytotoxicity against human cells. Results showed that these peptides were active against Gram-positive bacteria, Mycobacterium and fungi, but not against Gram-negative bacteria. On the other hand, these peptides show a slight toxicity towards nucleated human cells. In addition, the mechanism of action of ENN and BEA explaining their antibacterial action was studied.
The manuscript can be accepted for publication on “Toxins” considering the following points:
ENN is an abbreviation? Clarify the first time this acronym is reported; In table 1 standard deviation is missing; Same request for table 3; Several legends of figures report the software used, the authors should shift this information into materials and methods; Some names of species in the bibliography are not in italics: Lack of information on funds and Acknowledgments.
Finally, the authors should improve English.
Author Response
Dear Editor, Dear Reviewer,
First we would like to thank you for giving us the opportunity to submit a revised version of our work. We would like also to thank you for your helpful comments and suggestions that helped us to improve the quality of our work. We hope our answers will satisfy the reviewer. Please find below the list of questions and our answers to them. regards
Reviewer 2’s comments :
Comment 1 : The manuscript reports the biological characterization of five fungal cyclohexadepsipeptides. In particular their evaluated the antimicrobial activity and cytotoxicity against human cells. Results showed that these peptides were active against Gram-positive bacteria, Mycobacterium and fungi, but not against Gram-negative bacteria. On the other hand, these peptides show a slight toxicity towards nucleated human cells. In addition, the mechanism of action of ENN and BEA explaining their antibacterial action was studied.The manuscript can be accepted for publication on “Toxins” considering the following points:
ENN is an abbreviation? Clarify the first time this acronym is reported.
Answer to comment 1 : We would like to thank the reviewer for this suggestion. Accordingly, we added the abbreviation « ENN » when the term Enniatin is used for the first time in the text (i.e. in the abstract) in the revised version of the manuscript.
Comment 2 : In table 1 standard deviation is missing; Same request for table 3.
Answer to comment 2 : We thanks the reviewer for her/his question. In Table 1, values correspond to MIC values. MIC testing is performed three times but data are not expressed as means +/- SD. MIC have to be expressed as interval of value if different values are obtained for a particular peptide on a particular bacteria or as one value if the three assays give the same value of MIC for the peptide on that bacteria. This is the way MIC have to be expressed and are expressed in all published works (not only our works). For Table 3, the values given in this table correspond to the critical pressures of insertion determined for each peptide in each lipid. The critical pressure of insertion are determined from the slope’ equations given in Figure 4 considering X=0 in the equation. Due to that, only one value is given for each critical pressure of insertion of a particular peptide in a particular lipid. We could not calculate means +/- SD as only one value is obtain in each case.
Comment 3 : Several legends of figures report the software used, the authors should shift this information into materials and methods.
Answer to comment 3 : Accordingly to reviewer’s comment, we removed the mention of the software in the legends and shift it to materials and methods.
Comment 4 : Some names of species in the bibliography are not in italics.
Answer to comment 4 : Accordingly to reviewer’s comment, we putted all species names in italic in the bibliography.
Comment 5 : Lack of information on funds and Acknowledgments.
Answer to comment 5 : Accordingly to reviewer’s comment, we added these missing informations in the revised version of the manuscript.
Comment 6 : Finally, the authors should improve English.
Answer to comment 6 : Accordingly to reviewer’s comment, we asked the help of two english natives to correct the revised manuscript and added the following sentence in the acknowledgments: “We also would like to thank Mr Keith Cardozo (from Ireland) and Dr Adam James Mulkern from Aberystwyth University (UK) for their help during the editing of the manuscript.